# Assessment of Palm Jumeirah Island's Construction Effects on the Surrounding Water Quality and Surface Temperatures during 2001–2020

**Mohammad Mansourmoghaddam** [1], **Hamid Reza Ghafarian Malamiri** [1,2], **Iman Rousta** [2,3,4,*], **Haraldur Olafsson** [3,4] **and Hao Zhang** [5,*]

1   Department of Remote Sensing, Yazd University, Yazd 8915818411, Iran; mohammadmoghaddam@stu.yazd.ac.ir (M.M.); hrghafarian@yazd.ac.ir (H.R.G.M.)
2   Department of Geography, Yazd University, Yazd 8915818411, Iran
3   Institute for Atmospheric Sciences-Weather and Climate, University of Iceland and Icelandic Meteorological Office (IMO), Bustadavegur 7, IS-108 Reykjavik, Iceland; haraldur@vedur.is
4   Department of Physics, University of Iceland, Bustadavegur 7, IS-108 Reykjavik, Iceland
5   Department of Environmental Science, Engineering Jiangwan Campus, Fudan University, 2005 Songhu Road, Shanghai 200438, China
*   Correspondence: irousta@yazd.ac.ir (I.R.); zhanghao_fdu@fudan.edu.cn (H.Z.)

**Abstract:** Climate change stressors like rising and warmer seas, increased storms and droughts, and acidifying oceans are rapidly threatening coastal zones, which are the world's most densely inhabited places. This research assesses the effects of Palm Jumeirah Island (PJI) construction on its surrounding water quality and temperature, using Landsat-7 and 8 spectral and thermal bands for the years 2001, 2014, 2016, 2019, and 2020. To aid in this goal, the changes in water spectral reflectance was observed and interpreted, based on previous research and measurements, to discover the correlation between water quality and its spectral reflectance. Then, the sea surface temperature (SST) was calculated for the years under review and changes in water temperature were evaluated. Finally, the Green Normalized Difference Vegetation Index (GNDVI) and the Normalized Difference Turbidity Index (NDTI) were calculated to estimate water chlorophyll levels and water turbidity, respectively, and changes were observed and interpreted for the time period under review. The present study showed that the PJI construction not only increased the water reflectance in the 0.5–0.8 μm of wavelength, which can be considered to be the increase of suspended sediments and chlorophyll but the water temperature also increased by 7.5 °C during the 19 years. In addition, a gradual increase in the values of GNDVI (by 0.097–0.129) and NDTI (by 0.118~0.172) were observed. A drop in chlorophyll and suspended sediment spectral reflectance and GNDVI and NDTI values were also observed in 2020 compared to 2019 which can be attributed to the 63 to 82% decrease in tourists in Dubai in 2020 as a result of the COVID-19 pandemic. This study aims to draw attention to environmental issues by clarifying the effect of creating artificial islands in the sea and our analysis and results are a suitable reference for specialized hydrological and environmental studies based on spectral information and distance measurements, as presented in this paper.

**Keywords:** water pollution; SST; GNDVI; NDTI; Landsat images; human activities; remote sensing

## 1. Introduction

Coastal zones, which are the world's most densely populated locations, are quickly being threatened by climate change stressors such as rising and warmer seas, greater storms, and droughts [1], and acidifying oceans. Despite the fact that local human activities have long had an impact on coastal zones, it is still unknown how local human effects and climate change stressors may combine to endanger coastal ecosystems [2]. Coastal areas, as well as the food they produce, provide a significant contribution to the national economy and food

supply in the United Arab Emirates. In recent years, the substitution of shrimp farming for agriculture in coastal areas has been a paradigm shift, owing to increased shrimp production and changes in land use/land cover [3–5]. Human interventions, such as establishing shrimp [6] farms, not caring for drained swamps [7], aquaculture, commercial navigation, or using areas as sewage and industrial waste disposal sites [8], have caused an increase in the salinity of water near the shores [9]. The loss of soil fertility, a lack of fresh water for drinking, and the increased risk to public health are all consequences of seawater salinity (particularly near coasts) [6,10–13]. Because a rise in these pollutants might disrupt the ecology and harm human health and aquatic creature habitats, for those who live on beaches near the ocean, monitoring water-soluble chemicals (including water salinity) is crucial [9,14–16].

Assessing water quality resources is necessary for water purification. Water quality considers a body of water's thermal, physical, chemical, and biological qualities. It is a metric for determining whether or not water is fit for human consumption or recreational usage. Water quality is difficult to define since its use varies so much. For example, water characteristics that are acceptable in agriculture irrigation, differ from those which make water fit for human consumption [17]. Furthermore, water quality is frequently connected to consumer safety. Drinking water can include a broad range of chemical and microbiological pollutants, some of which can have significant health consequences for users. Understanding the types of contaminants that might enter the water supply and how they get there, is critical to ensuring water safety [17]. Water quality is also linked to discharges from urban, agricultural, and industrial sources. In addition, fecal contamination of surface water is a worldwide hazard caused by urban waste flows, and urban stormwater runoff has been demonstrated to contribute to surface water quality by harboring substantial amounts of fecal bacteria [18].

Traditional methods for measuring saline levels are both costly and time-consuming, requiring extensive primary data collection and laboratory testing [19–22]. So, the remote sensing approach, which has previously been demonstrated to be an effective way of monitoring water quality, might be an alternate tool to make the monitoring process more comfortable and successful [22–26]. Furthermore, Landsat data sets provide diverse satellite images, making monitoring and change detection easier and more efficient than primary data collection. In this respect, water's optical properties are relevant because they are affected by a variety of factors, including the concentration and features of suspended particles, dissolved solids, and other organic materials [27–29]. An effective monitoring strategy might be utilized based on the link between the optical features of water and the band values of the Landsat data series [19,29,30].

For various places, several investigators from across the world have developed many approaches for detecting water quality metrics using Landsat images [9,26,29,31–35]. Gonzalez-Marquez et al. (2018) demonstrated that Landsat-8 Operational Land Imager (OLI) images may be used to analyze water quality metrics such as phosphate concentrations, electrical conductivity, total suspended particles, turbidity, and pH in Mexico's coastal zones. In Iraq's Al-Huwaizah wetland, Landsat-8 OLI imagery was also shown to be capable of monitoring salinity. In another study, to build a monitoring strategy for the Mekong delta of Vietnam, Vu et al. (2018) focused on the link between in-situ salinity level data and Landsat-8 (OLI) band values [36]. Water quality metrics, such as suspended sediments, turbidity, chlorophyll-a, and others, may be mapped using Landsat-5 TM. To analyze the association between band values and field level parameter values, Nas et al. (2010) created regression models for these parameters using 28 band compositions [9].

Bugnot et al. (2018) used Landsat-5 (TM) and 7 (ETM+) to monitor and analyze environmental changes in Australian estuaries at the watershed level. Another water quality measure that may be determined using Landsat images is suspended solid concentration (SSC). To identify SSC from Landsat pictures, Shahzad et al. (2018) and Montaner et al. (2014) built empirical models. According to Chang et al. (2017), remote sensing techniques may be the most practical method of monitoring and regulating water quality [37]. Tur-

bidity, total suspended particles, and heavy metals such as iron, zinc, copper, chromium, lead, and cadmium may all be detected using Landsat spectral band data [9,21]. It may also be used to measure chlorophyll-a levels and water clarity. Using linear regression analysis, Vignolo et al. (2006) created a model to detect water quality index from Landsat-7 ETM band data, notably the blue and green bands. This research also shows that Landsat pictures' blue and green bands may be used to accurately determine water quality [38].

Morshed et al. (2016) used Landsat-7 ETM+ data to create a regression equation to identify soil salinity in coastal Bangladesh. Ferdous and Rahman (2018) likewise looked at the use of Landsat-5 TM to identify the region's soil water content [39]. Ferdous et al. (2019) have advocated using Landsat-8 OLI pictures to monitor the content of water using the total dissolved solids (TDS) index in surface water in coastal Bangladesh [40]. As a result of beach closures, reduced fishing, or deterioration of drinking water supplies, water quality has a significant impact on both public health and the economy [11]. Agricultural run-off and industrial wastewater, on the other hand, contribute to eutrophication processes, resulting in phytoplankton buildup. The increase in nutrient levels encourages excessive plant growth, which causes the water to become murky, has an impact on fish populations, and speeds up algal blooms [41,42]. Consideringly, this study aims to study the effect of Palm Jumeirah Island (PJI) construction in the Persian Gulf on the surrounding waters, to give a scientific view of the extent of changes in the water profile arising from the construction of the island. Since thermal remote sensing is a helpful technology for detecting thermal changes in marine systems that might alter the biological production rate [43,44], this study assesses the changes in PJI's surrounding water temperature during the 19 years from before the island existed, to its completion and subsequent growth. Additionally, by examining the correlation between parameters calculated from the characteristics of the water, the effect or effectiveness of each parameter, on/from other parameters, are shown. This study can be used as a scientific reference of water features for hydrologists conducting research in this area in the future, and also to monitor the trend of water changes during PJI's construction.

## 2. Materials and Methods

### 2.1. Study Area

The Study area is the PJI located on the Persian Gulf islands around Dubai and its surrounding waters. PJI is located on 25°07′09.2″ N 55°07′49.3″ E of geographic longitude and latitude coordinates (Figure 1). It has an area of 7.6 km$^2$ [45] and homes a population of 78,000 [46]. The Palm Jumeirah is the first of a series of man-made islands off Dubai's shore; the project began in 2001 [47] and was completed in 2008, adding 56 km to the coastline [48]. The project involved adding 120 million cubic meters of sand and 7 million tons of rock to the sea [46]. An offshore crescent barrier enclosing the island, with a total length of 11 km, was built at the same time to defend the island from wave attacks [49].

### 2.2. Data Collection

For the present study, Landsat-7/Enhanced Theme Mapper Plus (ETM+) and Landsat-8/Operational Land Imager and the Thermal Infrared Sensor (OLI/TIRS) images in path 160 and row 043 were used. These images were selected for the years 2001, 2014, 2016, 2019, and 2020, based on the closest images and the presence of suitable images without error and cloud for processing. The images were derived from the United States Geological Survey (USGS) portal. More details about the images used are provided in Table 1. August is the hottest month of the year in Dubai; to meet the images criteria, we tried to use images for this month. A gap was unavoidable as no perfect-quality images were found from August, or in the months before or after, in the period from 2001 to 2013.

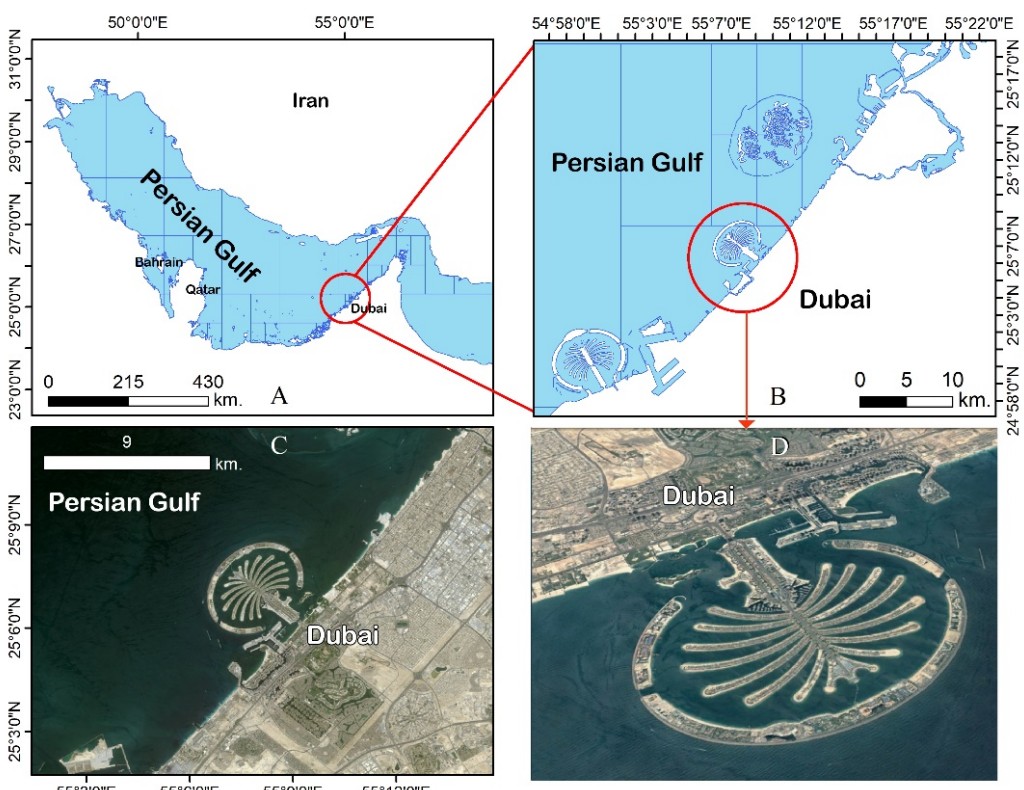

**Figure 1.** (**A**) Location of PJI, (**B**) location in zoom, (**C**) Landsat-8 RGB color image (2020), and (**D**) Google Earth image (2021).

**Table 1.** Details about the Landsat datasets used in this research.

| Sensor | Scene ID | Acquisition Date | Acquisition Time (GMT) | Cloud Cover (%) |
|---|---|---|---|---|
| Landsat-7 ETM+ | LE07_L1TP_160043_ 20010826_20200917_02_T1 | 26 September 2001 | 06:35:19 | 0 |
| Landsat-8 OLI/TIRS | LC08_L1TP_160043_ 20140822_20200911_02_T1 | 22 September 2014 | 06:46:38 | 0.5 |
| | LC08_L1TP_160043_ 20160827_20200906_02_T1 | 20 August 2016 | 06:46:43 | 0 |
| | LC08_L1TP_159035_ 20190914_20190917_01_T1 | 14 September 2019 | 06:46:43 | 0 |
| | LC08_L1TP_160043_ 20200806_20200916_02_T1 | 06 September 2020 | 06:46:32 | 2.8 |

### 2.3. Data Preprocessing

After collecting the images, they were corrected radiometrically and atmospherically through ENVI software and the Fast Line-of-sight Atmospheric Analysis of Spectral Hypercubes (FLAASH) algorithm, which is recommended for atmospheric correction over ocean or water body images. Then each image's spectral reflectance was analyzed through the transition of years from 2001 to 2014, 2016, 2019, and 2020. Then water indices were calculated to show and compare the changes in water content through changes in the indices values. Finally, the Sea Surface Temperature (SST) was calculated for the study period, to monitor changes in water temperature.

### 2.4. Derivation of Image Spectral Characteristics

To monitor water quality, remote sensing techniques rely on the capacity to measure changes caused by compounds in the spectral signature backscattered from water. Sus-

pended sediments (turbidity), thermal releases, algae (i.e., chlorophylls and carotenoids), dissolved organic matter, chemicals (i.e., nutrients, pesticides, and metals), pathogens, aquatic vascular plants, and oils are all major factors affecting water quality in water bodies across the landscape [50]. The energy spectrum of reflected solar and/or emitting thermal radiation from surface waters is changed by suspended sediments, algae, dissolved organic matter, oils, aquatic vascular plants, and thermal releases, which can be studied using remote sensing techniques [50]. Thus, by extracting the spectral behavior of the water around PJI in the studied time series, this study investigates the spectral reflectance changes to the waters of this region, over the studied years for each pixel. The pixels studied on water reflection as samples are as shown in Figure 2. The samples were selected in the same position for all years.

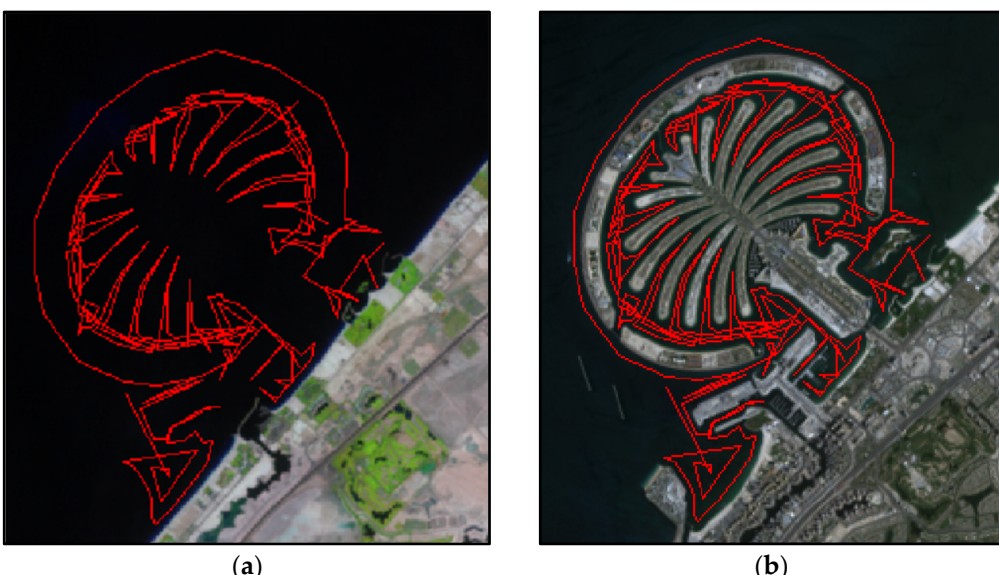

| (**a**) | (**b**) |

**Figure 2.** Study examples of changes in the spectral behavior of water around PJI (**a**) before PJI (2001), and (**b**) after PJI (in 2020).

*2.5. Calculation of the Sea Surface Temperature (SST)*

2.5.1. Calculation of the SST for ETM+ Sensor Single-Band

In order to calculate the SST from the ETM+ sensor thermal band (6th band), first of all, the DN (Digital Number) image of the thermal band converted radiance was calculated using Equation (1):

$$L_\lambda = ((L_{MAX_\lambda} - L_{MIN_\lambda})/(DN_{MAX} - DN_{MIN})) * (DN - DN_{MIN}) + L_{MIN_\lambda} \quad (1)$$

where $L_\lambda$ is spectral radiance (watts/m$^2$·str·µm)), $L_{MAX_\lambda}$ is spectral radiance which is correlated with DNMAX (watts/m$^2$·str·µm)), $L_{MIN_\lambda}$ is spectral radiance, which is correlated with DNMIN, and $DN_{MIN}$ is the minimum value of DN (1 or 0 based on LPGS or NLAPS Product, respectively). Then, an atmospheric correction was performed using the thermal atmospheric correction algorithm [51,52] in ENVI software. Then, thermal bands were converted to the effective temperature value using Equation (2):

$$T_{Landsat7} = K2/ln((K1/L_\lambda) + 1) - 273.15 \quad (2)$$

where $T_{Landsat7}$ is the effective temperature (°C), *K2* and *K1* are the calibration constants 2 and 1 for the thermal band Table 2. The thermal band used in this study is band 62 of Landsat-7, which is recommended as the more effective band for the SST calculation [53].

**Table 2.** The calibration constants of the ETM+ thermal band.

| Symbol | Constant (watts/m$^2$·str·µm) |
|--------|-------------------------------|
| $K1$ | 1282.71 |
| $K2$ | 1282.71 |

2.5.2. Calculating the SST for the TIRS Multi-Band

For the 11µm and 12µm channels, the multi-channel SST (MCSST) algorithm has been employed most often for SST retrieval from satellite data [54–61]. Although the single-window method was used in previous studies [62–64], the multi-window method has also been used in several studies [65–73]. To calculate the SST for the TIRS images (for the years 2014, 2016, 2019, and 2020) Equation (3) was used [73]:

$$SST = a_1 T_{11} + a_2 (T_{11} - T_{12}) + a_3 \tag{3}$$

where $a_1$, $a_2$, $a_3$ are the regression coefficients given in Table 3 and $T_{11}$ and $T_{12}$ are brightness temperature at 11 µm and 12 µm (in Celsius) which are the 10th and 11th bands of Landsat-8 (TIRS satellite).

**Table 3.** MCSST correlation coefficients [73].

| Symbol | Coefficient |
|--------|-------------|
| $a_1$ | 0.9767 |
| $a_2$ | 1.8362 |
| $a_3$ | 0.0699 |

Additionally, to ensure that the SST is not affected by other climatic variables, information on air temperature, wind speed, and relative humidity of the study area was extracted [74] on the day and hour of imaging and with the help of statistical parameters R2, *p*-Value, and Significance-F, significance between these parameters were obtained with the calculated SST for the study area.

*2.6. Calculation of Water Indices*

2.6.1. The Green Normalized Difference Vegetation Index (GNDVI)

The present study used the GNDVI index to evaluate the changes in the chlorophyll content of the water because the index is extremely sensitive to changes in chlorophyll content, which is inversely proportional to the water nitrogen content [75–77]. The GNDVI maps were calculated using Equation (4) [75]:

$$GNDVI = \frac{NIR - G}{NIR + G} \tag{4}$$

where $NIR$ is the near-infrared band which is the Landsat-8 (OLI) 5th and the Landsat-7 (ETM+) 4th bands and $G$ is the band with a 0.54~0.57 µm wavelength range which is the green band (Landsat-8 (OLI) 3rd and Landsat-7 (ETM+) 2nd bands).

2.6.2. The Normalized Difference Turbidity Index (NDTI)

The NDTI index was calculated for using remote sensing data to evaluate the turbidity of water in ponds and inland waters [77,78]. The NDTI maps were calculated using Equation (5) [78]:

$$GNDVI = \frac{R - G}{R + G} \tag{5}$$

where $R$ is the red band which is the Landsat-8 (OLI) 4th and the Landsat-7 (ETM+) 3rd bands.

### 2.6.3. Statistical Analysis

In order to statistically evaluate the parameters and indicators used in this study, using data matrices, the relationship and significance of the indicators over time per pixel were calculated using correlation coefficients and their $R^2$ using Microsoft excel software.

Figure 3 schematically shows the research method.

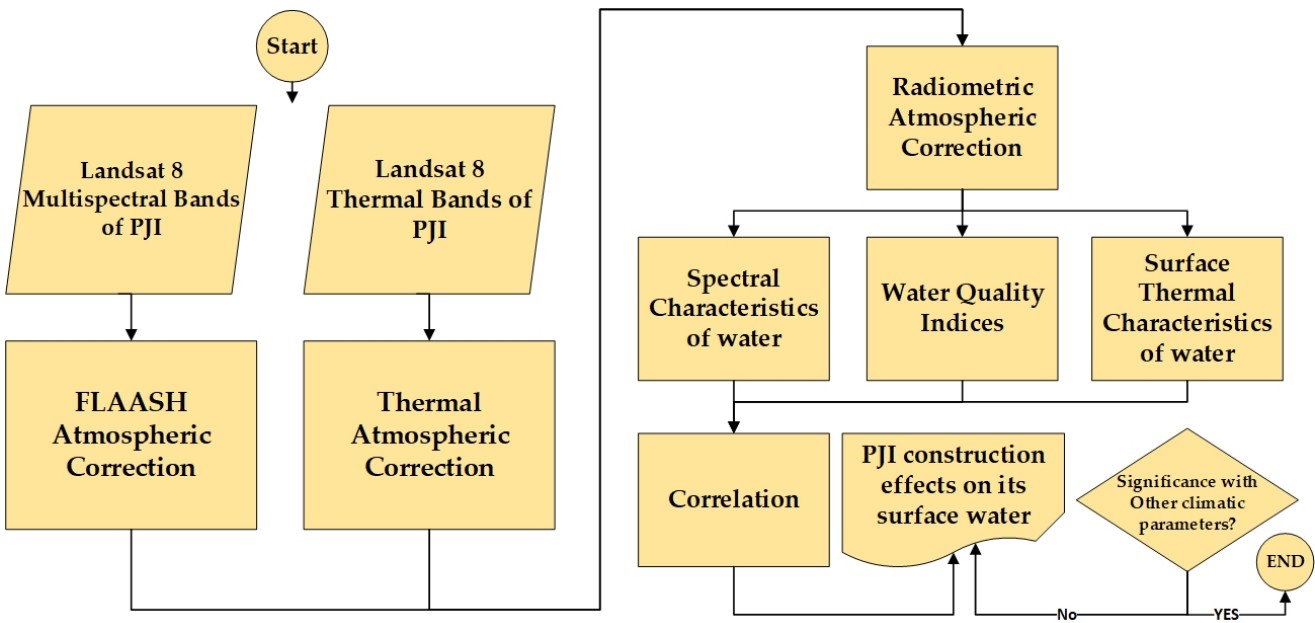

**Figure 3.** The flowcharts of the present research methods.

## 3. Results and Discussion

### 3.1. Changes in Water Spectral Characteristics

The study aims to evaluate the effects of island construction, specially PJI, Dubai, on its surrounding water reflectance, content, and temperature using remote sensing data to observe the changes during the years 2001 (before there was the island), 2014, 2016, 2019, and 2020. The results indicate that the water reflectance of the water around the island has been changed during the study period (Figure 4).

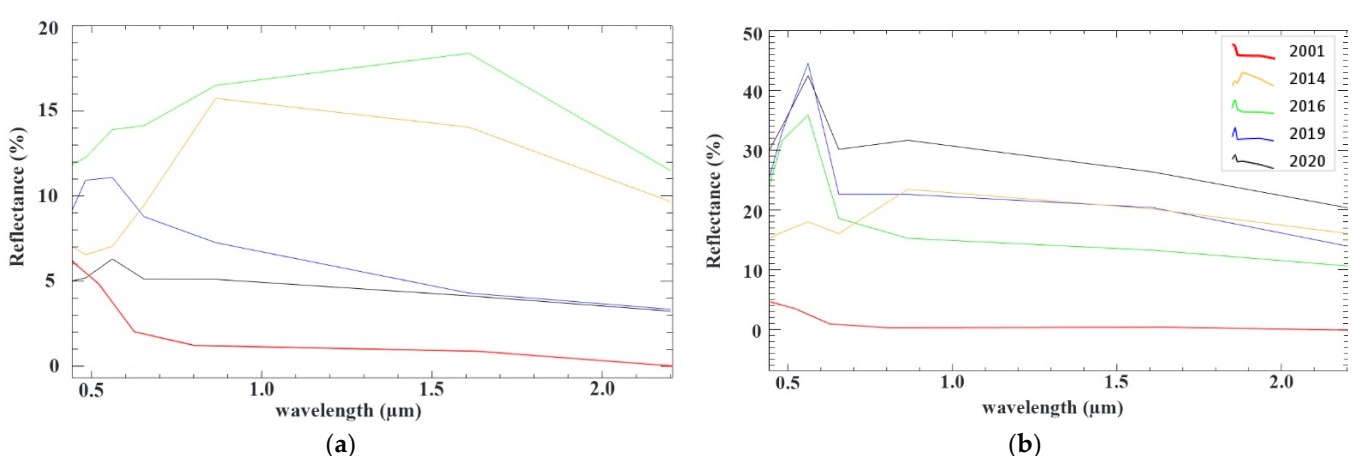

**Figure 4.** (**a**) Maximum and (**b**) mean reflectance of water surrounding PJI for the years 2001, 2014, 2016, 2019, and 2020.

Figure 4a clearly illustrates that the maximum reflectance has shifted to shorter wavelengths during the time period under review. The maximum water reflectance was

0.8~0.9 μm in 2014, and it had shifted to 0.5~0.6 μm in 2019 and 2020. Although the maximum reflectance of surrounding water was 1.6~1.7 μm in 2016, two peaks in 0.8~0.9 μm (near to 2014) and a high value of reflectance in 0.5~0.6 μm (near 2019 and 2020) are observed. It is while the reflectance was at 0.483 μm in 2001 that it shows an observable shift of the maximum reflectance to shorter wavelengths. Figure 4b also shows an almost regular trend with a regular increase of mean of water reflectance in 0.5~0.6 μm and 0.6~0.7 μm of wavelength from 2001 to 2016.

The comparison of the maximum water reflectance between 2014 and 2016 (Figure 4a) also clearly shows an increase in 0.8~0.9 μm in 2016, and the same increase has accrued between 2019 and 2020 in 0.5~0.6 μm of wavelength; the difference is that the maximum reflectance of 2019, in 0.5~0.6 μm, was much higher than in 2020. In the same trend as Figure 4a, the mean of water reflectance in 0.5~0.6 μm of wavelength was higher in 2019 than in 2020 (Figure 4b). This trend is significant according to Figure 5 in the perception of the amount of substances in water.

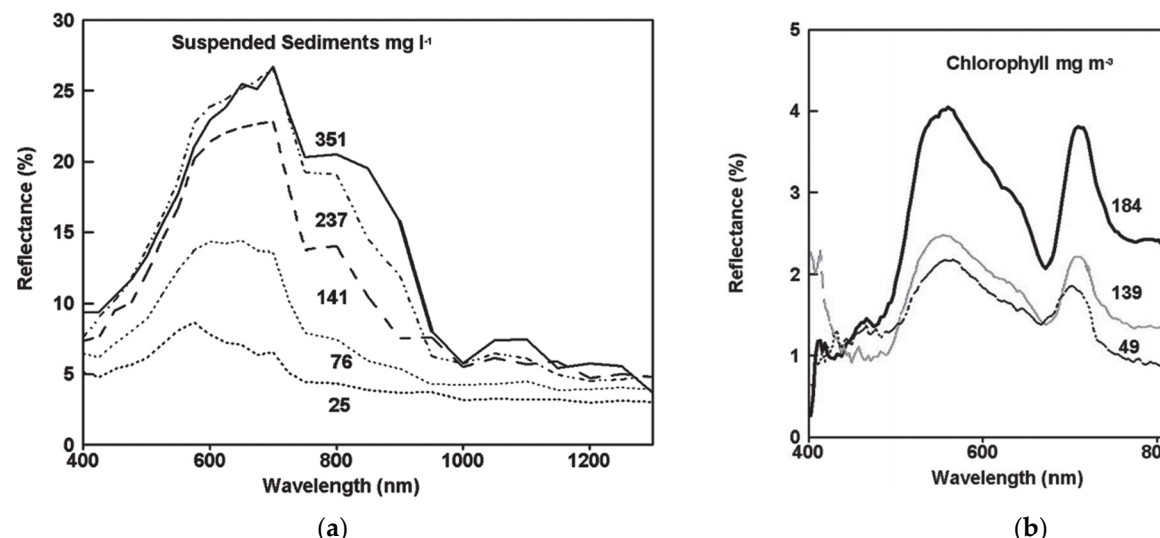

**Figure 5.** The relationship between reflectance and wavelength as affected by the (**a**) concentration of suspended sediments [43] and (**b**) chlorophyll concentrations [79].

### 3.2. Water Temperature Changes

Remote sensing mapping of absolute temperatures reveals geographical and temporal patterns of thermal releases that may be used to manage thermal releases. In this study, the assessment of water temperature was performed using SST algorithms. Thus, the SST maps for 2001, 2014, 2016, 2019, and 2020, and the overall changes to the maps, were calculated from ETM+ and TIRS thermal bands as shown in Figure 6.

As shown in Figure 6, the island's surrounding water surface temperature has gradually warmed since the island was built in 2001. The change map shows this change visually (Figure 6). The statistics of the SST maps indicate that the mean of the SST around the island has continuously increased from 27.5 °C in 2001 to 34.5 °C in 2014, 34.9 °C in 2016 and 2019, and finally, 35 °C in 2020. The maximum and minimum SST also increased constantly from 28 and 26.7 °C, respectively, in 2001 to 36.7 and 34.3 °C, respectively, in 2020 (Figure 7).

The results also show that there was no significance between the SST obtained in the study area and the climatic parameters of areas such as air temperature, wind speed, and relative humidity (Table 4).

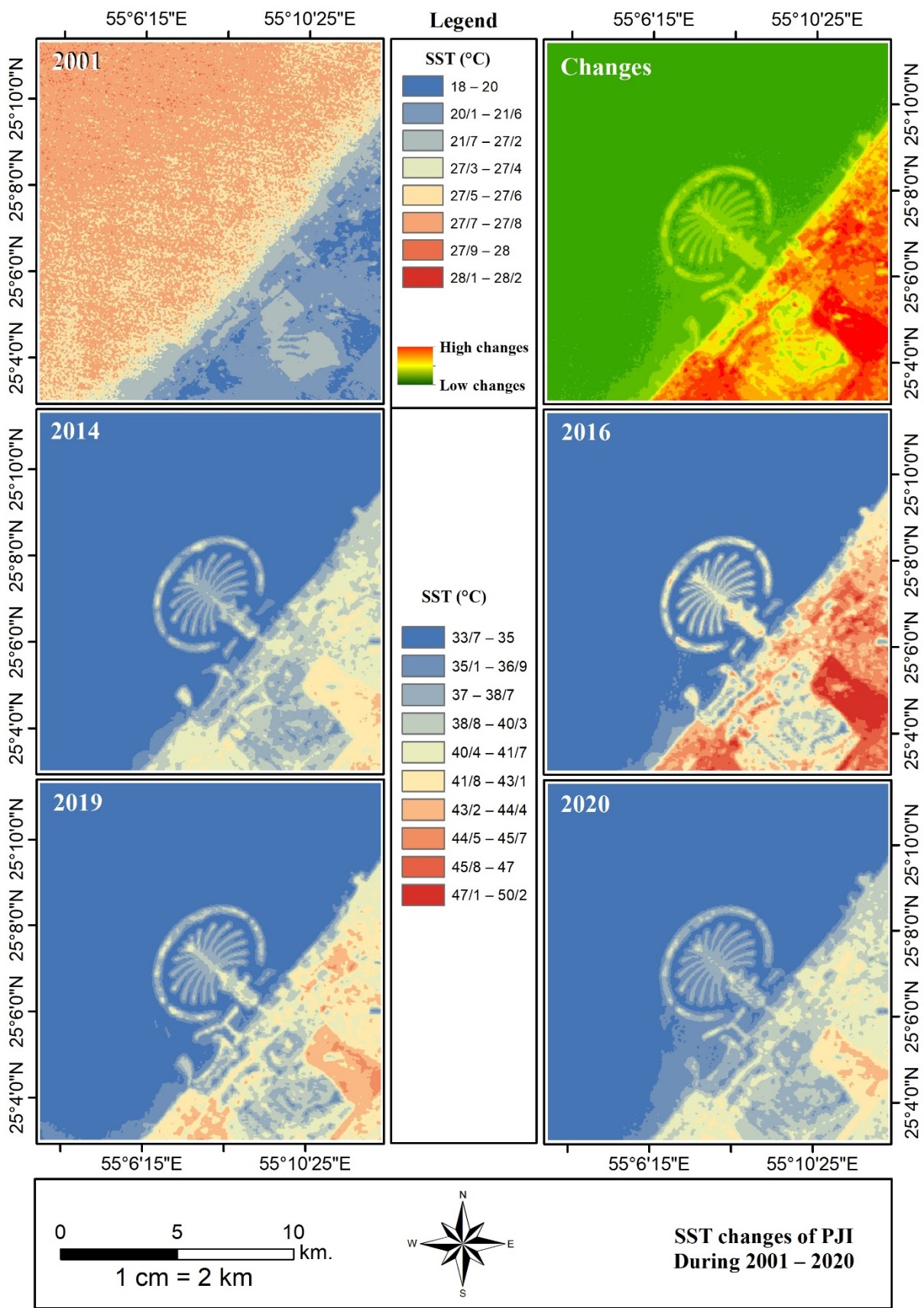

**Figure 6.** The calculated SST maps for 2001, 2014, 2016, 2019, and 2020 of PJI together with the overall map changes during this period.

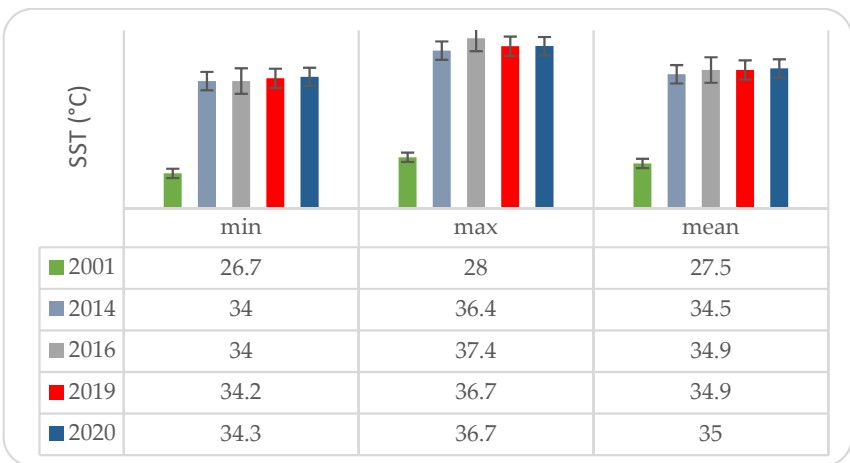

| | min | max | mean |
|---|---|---|---|
| ■ 2001 | 26.7 | 28 | 27.5 |
| ■ 2014 | 34 | 36.4 | 34.5 |
| ■ 2016 | 34 | 37.4 | 34.9 |
| ■ 2019 | 34.2 | 36.7 | 34.9 |
| ■ 2020 | 34.3 | 36.7 | 35 |

**Figure 7.** Statistics of the SST maps for 2001, 2014, 2016, 2019, and 2020 of PJI.

**Table 4.** Statistical values of the relationships between different climatic parameters and the SST in the study area.

| | Air Temperature | Wind Speed | Humidity |
|---|---|---|---|
| $R^2$ | 0/12 | 0/21 | 0/12 |
| *p*-Value | 0/65 | 0/54 | 0/71 |
| Significance F | 0/65 | 0/54 | 0/71 |

### 3.3. Assessment of Water Indices

Since many studies have used spectral indices in order to assess water quality, content, etc., in the present study we calculated some indices to find out the effects of the island's construction on the water quality in the PJI area. To achieve this goal, GNDVI and NDTI were calculated to give a clear view of the changes that have taken place in this region, during the study period. The maps of changes of these indices during the relevant time period are shown in Figure 8.

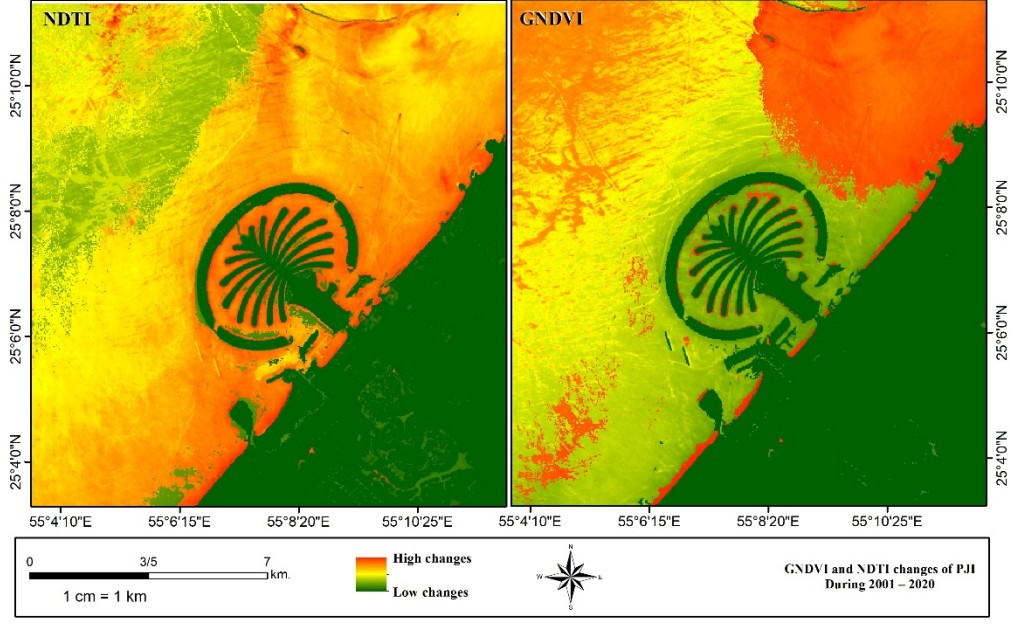

**Figure 8.** The GNDVI and NDTI change maps for PJI's surrounding water from 2001 to 2020.

The mean outputs of the GNDVI and NDTI calculations, which are more acceptable for deriving water parameters, are presented in Figure 9.

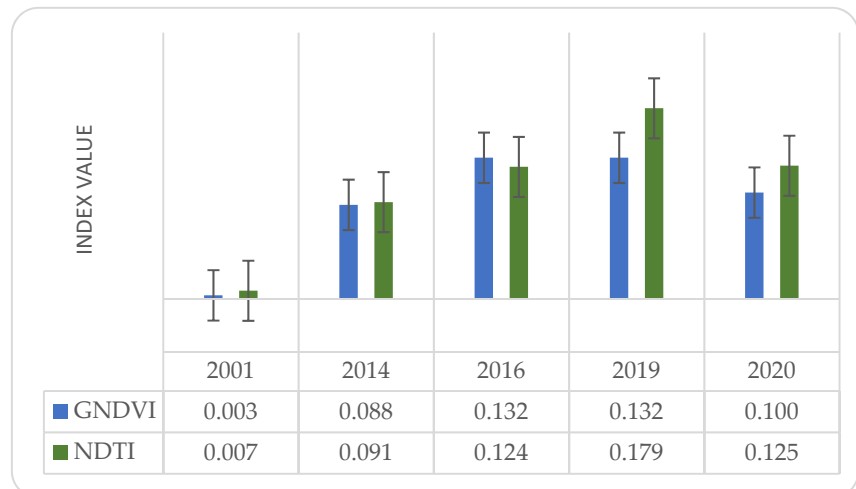

| | 2001 | 2014 | 2016 | 2019 | 2020 |
|---|---|---|---|---|---|
| ■ GNDVI | 0.003 | 0.088 | 0.132 | 0.132 | 0.100 |
| ■ NDTI | 0.007 | 0.091 | 0.124 | 0.179 | 0.125 |

**Figure 9.** Results of the GNDVI and NDTI calculations from PJI's surrounding water (mean).

Based on the indices results (Figure 9), GNDVI permanently increased from 0.003 in 2001 to 0.132 in 2019 in the water surrounding PJI. The NDTI results also confirm a similar trend around the island, it has had increased from 0.007 in 2001 to 0.179 in 2019. In 2020, a small decrease was observed in the GNDVI and NDTI results from 2019 (by 0.032 and 0.054, respectively) to 0.1 and 0.125, respectively.

The results show that increases in GNDVI, NDTI, and SST in the surrounding waters of PJI have been observed since building the island. However, the results of the statistical analysis of the matrix of changes in the pixels of the water area around PJI indicate that the highest significance and correlation was between the GNDVI results and the SST, with a correlation and $R^2$ of 85.3% and 80.76%, respectively. After this, the highest correlation and significance was observed between the NDTI results and the green band reflectance (GBR), with a correlation of 49.7% and an $R^2$ of 36.08%. Then came the correlation and significance between the GNDVI and NDTI results, with a correlation and $R^2$ of 39.1% and 44.81%, respectively, and NDTI and SST, with a correlation and $R^2$ of 34.5% and 40.77%, respectively. Finally, the least correlation and significance were obtained between GNDVI and GBR, with less than 20%, and SST and GBR, with less than 12% Table 5.

**Table 5.** Statistical relation between PJI water surrounding indices (sorted by descending).

| Rate | Symbol | $R^2$ (%) |
|---|---|---|
| 1 | GNDVI-SST | 80.76 * |
| 2 | NDTI-GBR | 36.08 |
| 3 | GNDVI-NDTI | 44.81 |
| 4 | NDTI-SST | 40.77 |
| 5 | GNDVI-GBR | 16.61 |
| 6 | SST-GBR | 11.7 |

Note: * Denotes significant at $p = 0.05$.

## 4. Discussion

The maximum reflectance has changed to shorter wavelengths over time, indicating that the maximum reflectance has altered. As a result, in 2014, the maximum water reflectance was 0.8~0.9 μm, whereas, in 2019 and 2020, it was 0.5~0.6 μm. Even though the highest depth of surrounding water was 1.61.7 m in 2016, two peaks in 0.8~0.9 μm (near 2014) and a high value of reflectance in 0.5~0.6 μm (near 2019 and 2020) have been detected. The results depict an observable shift of the maximum reflectance to shorter

wavelengths, while the reflectance was 0.483 μm in 2001. From 2001 to 2016, the results also demonstrate an extremely consistent trend of increasing mean water reflectance in the 0.5~0.6 and 0.6~0.7 μm wavelength ranges. Significant connections have been described between suspended sediments and brightness or reflectance measured on satellite sensors using spectral bands or combinations of bands before [43,50]. So the increase in PJI's surrounding water reflectance in 0.5~0.6 μm of wavelength, reveals the increase in the concentration of suspended sediments in the water due to the island's construction. Increased reflection at 0.5~0.65 μm wavelengths may also indicate an increase in chlorophyll [79] at the water surface. In addition, the maximum water reflectance between 2014 and 2016 (Figure 4a) indicates a definite rise in 0.8~0.9 μm in 2016 and a similar increase in 0.5~0.6 μm wavelength between 2019 and 2020. The difference is that the maximum reflectance of 2019 in 0.5~0.6 μm was significantly higher than in 2020 (Figure 4b). Considering that the inhabitants of this island are of seventy different nationalities [80], the COVID-19 lockdown has led to a decrease in travel and human activity in the region, so this reduction in the maximum reflectance may be due to a 68 to 82% fall in tourism to Dubai in 2020, due to the COVID-19 pandemic [81]. In this regard, many prior studies have proven the effect of lockdown in the COVID-19 era [82–84] and foreign direct investment [85] on environmental pollution.

According to the results, the SST of the waters around PJI has increased by a maximum of 27.2% over 19 years. While earlier Landsat satellite studies used a mono-window technique with single-channel data for SST retrieval [62–64], the Landsat 8 OLI/TIRS allowed for the use of a split-window method with multi-channel data for the SST estimate [65–73]. Thus, comparing the following years with 2001 (27.5 °C), in 2014, this temperature had a difference of 25.5% to 34.5 °C, in 2016 and 2019, a difference of 26.7% to 34.9 °C, and finally, in 2020, a difference of 27.2% to 35 °C. When human impact alters the temperature of a body of water, thermal pollution occurs. So, in addition to water reflectance, considering that no significant relationship was observed in the results between the SST values and the climatic variables of air temperature, wind speed, and relative humidity, the change in SST in the water surrounding PJI is clearly the effect of the island's construction. The results show that the increase in the GNDVI and NDTI was up to a maximum of 4203% and 2321%, respectively. This amount was recorded from 0.0003 in 2001, with a 2764.7% difference, with 0.0088 in 2014, with a 4203% difference, with 0.132 in 2016 and 2019, and with a 3139.2% difference, with 0.1 in 2020. These differences may be because the creation of the island has increased the water chlorophyll, as the indices are sensitive to chlorophyll content changes [76,77]. Further, as an estimation for water turbidity, the NDTI results clearly show an increase in turbidity in the water surrounding PJI. In 2020, both the GNDVI and NDTI decreased somewhat (by 0.032 and 0.054, respectively) to 0.1 and 0.125, respectively. This, as mentioned before, could be due to a decrease of 68% to 82% of tourists in Dubai in 2020.

In addition, the results obtained from the study with respect to the correlation and significance between indices (and GBR), indicate that the highest correlation and $R^2$ (above 80%) was between GNDVI and SST. This can be a good indication of the relationship between these two indicators and the effect of the increased GNDVI values (increasing water chlorophyll content) and the increased water surface temperature. Next, NDTI and GBR values, with a correlation coefficient of 49.7% and an $R^2$ of 36.08%, have a relatively high correlation and significance, which clearly indicates the relationship between these two parameters and the increase in the reflection in the green wavelength and the increase in the NDTI values during the study period and after the construction of PJI. The third significant correlation, with the correlation coefficient of 39.1% and $R^2$ of 44.81%, was between NDTI and GNDVI, which can indicate the effect of these two parameters on each other during the statistical period and the consequent increase in both of these parameters (increased chlorophyll and suspended sediments) after PJI's fabrication. In addition, NDTI and SST, with a correlation coefficient of 34.5% and $R^2$ of 40.77%, indicate a relatively significant relationship between these two parameters and could indicate that this index is similar to GNDVI but with a lower coefficient that has increased the water surface temperature.

Both GNDVI and SST values had a low correlation and significance (less than 20% and 12%, respectively) with GBR, indicating that these two parameters were not directly related to or affected by GBR.

## 5. Conclusions

This study aimed to investigate the effect of PJI's construction on the quality of its surrounding waters from 2001 (before island construction) to 2020. According to the results, the construction of this island has changed the spectral characteristics of the surrounding water and has increased water reflection in the range of 0.5 to 0.8 μm. This increase can be considered to be an increase in suspended particles and chlorophyll in the water. The construction also raised the average water temperature by 7.5 °C, over 19 years. The results obtained from the study of GNDI and NDTI also showed an increase in the values of these indices from 2001 to 2019, which can be considered to be an increase in the water's chlorophyll content and water turbidity. The study found a slight drop in the values of these indicators in 2020, which is likely due to a sharp decline (68 to 82%) in tourists in Dubai in 2020 (due to the COVID-19 pandemic). The study results were obtained using Landsat-7/ETM+ and Landsat-8 OLI/TIRS images to assess the quality and temperature of the seawater in the study area around the island. These results can be used as a reference for more specialized studies by experts in the field of hydrology, using the remote sensing data which is presented in this research.

**Author Contributions:** Conceptualization, M.M.; data curation, M.M. and H.R.G.M.; funding acquisition, H.Z.; methodology, M.M., H.R.G.M. and I.R.; software, M.M., H.R.G.M. and I.R.; supervision, H.R.G.M., H.O. and H.Z.; validation, H.R.G.M., H.O. and H.Z.; visualization, M.M.; Writing—review & editing, M.M. and H.Z. All authors have read and agreed to the published version of the manuscript.

**Funding:** This study is supported by Shanghai Municipal Science and Technology Commission within the international cooperation framework of the Youth Scientists from the 'One Belt and One Road' countries.

**Institutional Review Board Statement:** Not Applicable.

**Informed Consent Statement:** Not Applicable.

**Data Availability Statement:** Contact to mohammadmoghaddam@stu.yazd.ac.ir or irousta@yazd.ac.ir.

**Acknowledgments:** I.R. is deeply grateful to his supervisor (Haraldur Olafsson, Professor of Atmospheric Sciences, Institute for Atmospheric Sciences-Weather and Climate, and the Department of Physics, University of Iceland, and Icelandic Meteorological Office (IMO)), for his great support, kind guidance, and encouragement.

**Conflicts of Interest:** The authors declare no conflict of interest.

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
