# Peer review of "Assessment of Palm Jumeirah Island’s Construction Effects on the Surrounding Water Quality and Surface Temperatures during 2001–2020"

_water, doi:10.3390/w14040634_

Round 1

Reviewer 1 Report

Authors should revise the manuscript submitted. Several mistakes can be found indicating that part of the information/text has been unintentionally deleted :
Error! Reference source not found
It seems that most of the time is related to figures or tables references.
Please check it.
Materials and methods should be improved. The methodology explanation might be clarified
In figure 3, use reflectance instead of data value for y-axis title. Try to use a larger font size, at least similar to figure 4 to improve its quality and make it easier to read.
Please include stadistical errors of the data presented in result tables

Author Response

Dear Reviewer, Thank you for your attention and comments, the answer file to the questions and the modified version of the article will be attached.

Reviewer 2 Report

The paper is interesting, it is based on many observations, but, unfortunately, many errors appear in the text and in this form it cannot be appreciated and published. A massive revision of the text is needed.

You must check all bibliographic references. In many places, instead of a reference, the text “Error! Reference source not found. ”
Also, when indicating the bibliography of the figures, texts of the type appear, as e.g. line 175: The samples studied on water reflection are as shown in Error! Reference source not found.
The whole text needs to be revised and redone because the same text “Error! Reference source not found. ” it appears in many places and the value of the work cannot be clearly determined.

Author Response

(The authors gave the same response as above.)

Reviewer 3 Report

 The manuscript is written well and is a very elaborative study that contains data from 2001 to 2020. The work is suitable for the water journal. However, I have some general questions which need to be addressed before publication.  The title needs to be modified as authors basically study the effect of urbanization on the surrounding water quality.  Authors need to explain how they generated the spectral behavior data in figure 2 " Figure 2. Study examples of changes in the spectral behavior of water around PJI before PJI(2001, a) 177 and after PJI (in 2020, b) Similarly, author needs to elaborate regarding the measuring  suspended particles and chlorophyll in water

Author Response

(The authors gave the same response as above.)

Round 2

Reviewer 2 Report

I agree that the paper should be published.

Author Response

Thank you for your efforts, accuracy, and kindness, dear reviewer, Your comment made us happy and motivated.